# ControlSwap: Controllable Personalized Subject Swapping

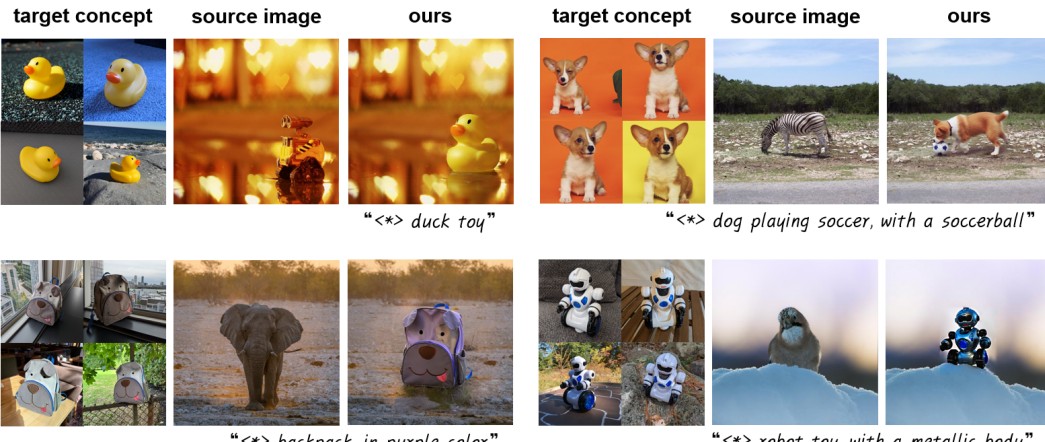

Figure 1: **ControlSwap** enables high-controllability personalized subject swapping. Compared to existing methods, our approach better adapts the target subject to new poses/contexts while preserving background fidelity.

## Abstract

Personalized subject swapping aims to replace a source concept in a source image with a user-specified target concept while preserving the rest of the scene. Despite recent advances, two key controllability issues persist: (1) object-agnostic foreground processing impedes identity transfer under pose and context changes; and (2) background regions that should remain unchanged are unintentionally altered. This lack of controllability profoundly curtails expressivity and narrows real-world editing applicability. We introduce ControlSwap, an SDXL-based framework that redesigns the refiner into an identity-preserving Identifier. In contrast to prior approaches that apply a uniform, object-agnostic loss across the entire image, ControlLoss decomposes optimization into object-specific and nonobject-specific objectives, enabling fine-grained, region-aware control. Experiments demonstrate improved controllability and background preservation across diverse personalization scenarios. Code and checkpoints will be released to support reproducibility and future research.

## 1 Introduction

Generative diffusion models (Ho et al., 2020; Song et al., 2021; Rombach et al., 2022) have significantly expanded the scope and quality, achieving state-of-the-art results in image synthesis, inpainting, and editing. Diffusion-based image editing (Saharia et al., 2022; Lugmayr et al., 2022; Meng et al., 2022; Hertz et al., 2023b; Brooks et al., 2023) enables fine-grained modifications to existing images while preserving realism.

Figure 2: Problem illustration: Existing personalized subject swapping methods often fail to adapt the target subject to new poses/contexts or unintentionally alter the background.

Within this broader editing landscape, personalized subject swapping—replacing a source concept in a source image with a user-specified target concept object while preserving the rest of the scene (Gu et al., 2023; 2024; Zhu et al., 2025)—has emerged as a challenging and practically valuable task. Yet, despite this progress, two limitations persist:

1. **Inversion pipelines and background drift.** DDIM-Inversion-based methods reconstruct the latent trajectory of the source image through inversion(Mokady et al., 2023; Ju et al., 2024; Gu et al., 2023; 2024). Due to the stochastic nature of diffusion, this process often fail to preserve background regions that should remain unchanged (Lin et al., 2024). Moreover, reliance on source masks (Gu et al., 2024) can amplify artifacts under source–target *shape mismatch*, resulting in composites that do not harmonize with the context.

2. **Object-agnostic treatment in the subject region.** Score-distillation-style approaches (Poole et al., 2023; Hertz et al., 2023a; Nam et al., 2024; Zhu et al., 2025) provide largely *object-agnostic guidance*, which hampers identity transfer when poses or contexts change—a central requirement in personalization (Ruiz et al., 2023).

This lack of controllability diminishes semantic expressivity while restricting the scope of practical editing tasks. Therefore, we frame personalized subject swapping through the lens of *controllability*: (i) within the subject region, faithfully adapt identity of target concept to novel poses and contexts; (ii) within the nonsubject context, avoid unintended edits.

We identify two underlying causes. First, many previous methods are built on SD-style architectures with a single encoder trained primarily at $512 \times 512$, which limits representational capacity (Podell et al., 2024). In contrast, we adopt SDXL as the baseline, leveraging its dual-encoder and redesigning the refiner into an identifier that preserves target concept identity. Second, existing objectives are applied uniformly over the object area, even though some pixels within the bounding box belong to the object core while others do not; this object-agnostic treatment increases sensitivity to the initial latent and harms controllability. Our region-aware *ControlLoss* removes this dependence by replacing identity on object-core pixels and enforces consistency on nonobject context and enabling natural integration into the scene. Object and nonobject regions are obtained from a segmentation mask (Kirillov et al., 2023).

For evaluation, we use SwapBench from ConSwapBench (Gu et al., 2024) as the source image benchmark. Since ConceptBench removes backgrounds—simplifying the distribution and understanding personalization difficulty— we instead draw subject identities from DreamBooth dataset (Ruiz et al., 2023). Both qualitative and quantitative results show that our method substantially improves controllability over baselines, enabling more accurate subject personalization while preserving background fidelity.

**Contributions.**

- **Problem perspective.** We formalize *controllability* as a dual requirement over personalization and preserving background regions for personalized subject swapping.
- **Architecture.** We redesign the SDXL-based pipeline around a dedicated identity-preserving refiner, *Identifiner*, which maintains subject identity while remaining prompt-faithful.
- **Region-aware objective.** We introduce a decoupled, mask-driven objective, *ControlLoss*, that alleviates dependence on the source latent. It treats the object core and the surrounding nonobject context separately, enabling fine-grained control without cross-region entanglement.
- **Experimental Results.** We used identities from DreamBooth and tested on SwapBench. Qualitative and quantitative results demonstrate consistent improvements in controllability—better personalization of the target concept and stronger fidelity of the preserved background—.

## 2 RELATED WORK

### 2.1 DIFFUSION-BASED IMAGE EDITING

With the advent of text-to-image diffusion models, image editing shifted from early GAN-based approaches (Karras et al., 2019), confined to narrow object domains and lacked generalization. Diffusion-based methods (Nichol et al., 2022; Brooks et al., 2023) overcome these limitations.

Training-based methods (Brooks et al., 2023; Sheynin et al., 2024) use instruction–image pairs to follow natural-language edits and flexibly alter the subject identity. At test time, inversion-style approaches (Mokady et al., 2023; Ju et al., 2024) recover the reverse diffusion trajectory of a source image. Moreover, training-free methods (Meng et al., 2022; Couairon et al., 2023) perform noise-based resampling and mask-guided local editing. Attention manipulation techniques (Hertz et al., 2023b; Cao et al., 2023) further enable fine-grained control. Finally, optimization-based methods (Poole et al., 2023; Hertz et al., 2023a; Nam et al., 2024) employ score-distillation objectives to optimize latent representations to support subject adaptation with scene-level consistency.

### 2.2 PERSONALIZED SUBJECT SWAPPING

Personalization methods (Ruiz et al., 2023; Gal et al., 2023; Kumari et al., 2023) use few-shot tuning, while editing methods (Li et al., 2023; Choi et al., 2023; Gu et al., 2023) integrate attention manipulation with personalized embeddings to better preserve identity. Paint-by-Example (Yang et al., 2023) first formulated exemplar-guided swapping by extracting exemplar features with a CLIP encoder and injecting them via cross-attention.

More recently, PhotoSwap (Gu et al., 2023) injects attention features from the DDIM-inverted source trajectory to preserve layout and background. As a follow-up, SwapAnything (Gu et al., 2024) introduces externally obtained segmentation masks (Kirillov et al., 2023) to improve object-level precision. However, inversion-based methods cannot maintain background well because of the nature of probabilistic nature of diffusion models. Thus, segmentation-based masks can cause artifacts from shape difference between source and target concept. Complementary efforts (Zhu et al., 2025) follow score-distillation trajectories for targeted swapping, aiming to balance identity transfer and scene preservation. Because DDS-based loss optimization depends on the source latent, it can limit editing freedom.

## 3 PRELIMINARIES

### 3.1 DIFFUSION MODELS AND SDXL

Let $\mathbf{x}_0 \in \mathbb{R}^{H \times W \times C}$ be a clean image and $\mathbf{x}_t$ its noised version at step $t$ under variance schedule $\{\beta_t\}_{t=1}^{T}$. The forward process incrementally adds Gaussian noise, and the reverse process is learned via a noise-prediction network $\boldsymbol{\epsilon}_\theta(\mathbf{x}_t, y, t)$ conditioned on a text prompt $y$, following DDPM (Ho et al., 2020):

$$p_\theta(\mathbf{x}_{t-1} \mid \mathbf{x}_t, y) \approx \mathcal{N}\left( \frac{1}{\sqrt{\alpha_t}} \left( \mathbf{x}_t - \frac{\beta_t}{\sqrt{1 - \bar{\alpha}_t}} \, \boldsymbol{\epsilon}_\theta(\mathbf{x}_t, y, t) \right), \, \sigma_t^2 \, \mathbf{I} \right), \quad (1)$$

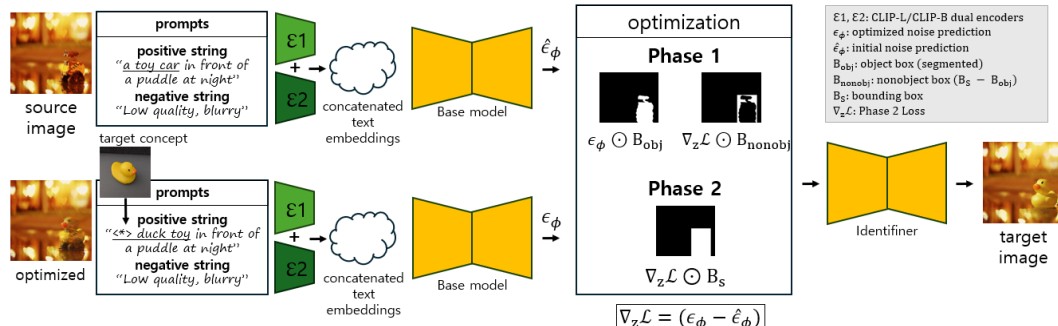

Figure 3: Overview of the proposed **ControlSwap** framework. Our method replaces the target subject in a source concept with a personalized reference object while preserving the background. It leverages SDXL with dual encoders and a redesigned *identifier* module for identity preservation. Region-aware loss scheduling by ControlLoss separately emphasizes object regions and their surrounding context during early optimization, and then gradually shifts focus toward consistent context refinement in later stages.

where $\sigma_t^2$ denotes the (possibly learned) variance at step $t$, where $\alpha_t = 1 - \beta_t$ and $\bar{\alpha}_t = \prod_{s=1}^{t} \alpha_s$.

**SDXL conditioning.** Stable Diffusion XL (SDXL) (Podell et al., 2024) extends SD by introducing dual text encoders—a large CLIP-L and a base CLIP-B—whose embeddings are concatenated and fed into the denoising UNet. Denoting the denoiser as $f_\theta$ and the text encoders as $E_{\text{CLIP-L}}$ and $E_{\text{CLIP-B}}$, the conditional score is

$$\boldsymbol{\epsilon}_\theta(\mathbf{x}_t, t, y) = f_\theta\big(\mathbf{x}_t, t, \; [E_{\text{CLIP-L}}(y) \,\|\, E_{\text{CLIP-B}}(y)]\big), \tag{2}$$

where $\|$ denotes concatenation. This dual-encoder design enriches semantic conditioning and supports $1024^2$ generation, making SDXL a strong backbone for subject-driven editing.

### 3.2 SCORE-BASED EDITING OBJECTIVES

Let $g_\phi$ be a differentiable generator (decoder) with latent parameter $z$ and $\mathbf{x}_0(z) = g_\phi(z)$. At timestep $t$, let $\mathbf{x}_t$ denote a noised sample of $\mathbf{x}_0(z)$ and $\boldsymbol{\epsilon} \sim \mathcal{N}(\mathbf{0}, \mathbf{I})$ the injected Gaussian noise.

**SDS.** Score Distillation Sampling (SDS) (Poole et al., 2023) aligns the generated image with a target prompt $y$ by matching predicted and true noise:

$$\nabla_z \mathcal{L}_{\text{SDS}}(z) = w(t) \left( \boldsymbol{\epsilon}_\theta(\mathbf{x}_t, y, t) - \boldsymbol{\epsilon} \right) \frac{\partial \mathbf{x}_0(z)}{\partial z}, \tag{3}$$

where $w(t)$ is a timestep-dependent weight.

**DDS.** Delta Denoising Score (DDS) emphasizes the semantic change between a target prompt $y_{\text{tgt}}$ and a source prompt $\hat{y}_{\text{src}}$ by differencing scores. Let $\hat{\mathbf{x}}_t$ be the noised source image; then

$$\nabla_z \mathcal{L}_{\text{DDS}}(z) = w(t) \left( \boldsymbol{\epsilon}_\theta\big(\mathbf{x}_t, y_{\text{tgt}}, t\big) - \boldsymbol{\epsilon}_\theta\big(\hat{\mathbf{x}}_t, \hat{y}_{\text{src}}, t\big) \right) \frac{\partial \mathbf{x}_0(z)}{\partial z}, \tag{4}$$

which suppresses noisy content and emphasizes desired direction (Hertz et al., 2023a; Nam et al., 2024).

## 4 CONTROLSWAP

### 4.1 OVERALL PIPELINE

ControlSwap is a controllable personalized subject swapping framework built on SDXL to overcome the controllability limits of existing methods. Figure 3 provides overview of our framework:

given a source image and a target prompt, SDXL performs latent optimization under each phase of object-aware objective ControlLoss, and the identifiner refines the subject identity while keeping the background fixed. We introduce two core ideas: (i) an SDXL-based architecture with a redesigned identity-preserving *Identifiner*; and (ii) a region-aware optimization strategy *ControlLoss* that decouples object transformation from background preservation. The remainder of this section details the architecture (Sec. 4.2) and the region-aware objective ControlSwap (Sec. 4.3).

## 4.2 SDXL Backbone and Identifiner

Most prior subject swapping use Stable Diffusion backbones, whose single encoder and limited parameter count restrict controllability. We instead adopt SDXL for its dual-encoder design and larger capacity. Target concept is trained by DreamBooth-LoRA (Ruiz et al., 2023; Hu et al., 2022) to capture concept identity while generalizing across contexts. However, as shown in Figure 7, stronger refinement of vanilla refiner degrades subject identity and even alters the background. This occurs because the refiner is not trained to preserve target concepts. To address this, we propose the *Identifiner*, which reuses the SDXL base model in an inpainting formulation to restrict denoising to foreground regions while freezing the background.

**Identifiner Redesign** Let $B_S \in \{0,1\}^{H \times W}$ be the bounding box (1 for subject pixels, 0 otherwise). We first encode the background latents and keep them fixed:

$$z^{\text{bg}} = \mathcal{E}(x).$$

At each denoising step $t$ of the Identifiner, only the subject latents are updated:

$$\tilde{z}_{t-1} = \epsilon(z_t, y; t), \qquad \tilde{z}_{t-1} = z_{t-1} \odot B_S.$$

Finally, we decode the composite target image $x_{\text{target}}$:

$$x_{\text{target}} = \mathcal{D}(z_0).$$

This masked refinement ensures that identity is preserved and enhanced strictly within $B_S$, while the background remains unchanged.

## 4.3 ControlLoss Scheduling

Our optimization process is divided into two phases, illustrated in Figure 3. We obtain segmentation masks (Kirillov et al., 2023) for object and nonobject foreground regions. Let $B_{\text{obj}}$ denote the binary mask for the object region, $B_{\text{nonobj}}$ the foreground nonobject region (e.g., background elements overlapping the bounding box), and $B_S$ bounding box in the target image.

**Phase 1: Object and Nonobject Separation.** In the early optimization stage, the primary goal is to allow the target subject to deviate freely from the source object in order to capture new identity features. To this end, we define separate region-wise terms: an object-core loss $\mathcal{L}_{\text{obj}}$ that drives semantic transformation of the subject, and a near-context loss $\mathcal{L}_{\text{nonobj}}$ that constrains local surroundings to prevent spill-over of edits. Explicitly:

$$\mathcal{L}_{\text{obj}} = \epsilon_\phi - \epsilon, \tag{5}$$

$$\mathcal{L}_{\text{nonobj}} = \left(\epsilon_\phi - \epsilon\right) - \left(\hat{\epsilon}_\phi - \epsilon\right). \tag{6}$$

The combined pixel-wise **ControlLoss** is then applied as

$$\nabla_z \mathcal{L}_{\text{Control}} = \begin{cases} \mathcal{L}_{\text{obj}}, & \text{if } (x,y) \in B_{\text{obj}}, \\ \mathcal{L}_{\text{nonobj}}, & \text{if } (x,y) \in B_{\text{nonobj}}, \\ 0, & \text{otherwise (background).} \end{cases} \tag{7}$$

This formulation ensures that object pixels are encouraged to transform toward the target semantics, while nonobject foreground pixels act as a buffer region that aligns with both source and target, thus stabilizing context without over-editing the background.

**Phase 2: Subject Integration.** Once the target subject has acquired the desired identity and pose, the optimization shifts to refining its integration into the scene. In this stage, the focus is on enforcing coherence between the subject and its surroundings. Therefore, the ControlLoss reduces to a consistency term over the subject mask $B_S$:

$$\mathcal{L}_{\text{fore}} = (\epsilon_\phi - \epsilon) - (\hat{\epsilon}_\phi - \epsilon). \tag{8}$$

$$\nabla_z \mathcal{L}_{\text{Control}} = \begin{cases} \mathcal{L}_{\text{fore}}, & \text{if } (x, y) \in B_S, \\ 0, & \text{otherwise (background)}. \end{cases} \tag{9}$$

This phase enforces identity-context consistency, improving visual harmony and preventing unnatural seams at subject boundaries, while guaranteeing that background regions stay intact.

## 5 EXPERIMENTS

We evaluate *ControlSwap* both qualitatively and quantitatively. To enable more challenging personalization, we selected 8 target concepts and trained using DreamBooth. For quantitative evaluation, we leveraged SwapBench from ConSwapBench, containing 160 source images and bounding boxes. Positive prompts were recommended by LLM model to reflect a wide range of personalization scenarios. This setup evaluates personalized subject swapping.

### 5.1 QUALITATIVE RESULTS

Figure 4 presents a visual comparison between our method and existing approaches (Poole et al., 2023; Hertz et al., 2023a; Nam et al., 2024; Zhu et al., 2025; Gu et al., 2023; 2024; Ju et al., 2024; Hertz et al., 2023b). Existing methods often struggle to achieve controllability: SDS over-smooths or distort the subject, undermining identity fidelity; DDS fails to control both target concept identity and prompt fidelity; CDS and InstantSwap fail to adapt the subject to new poses or contexts; PhotoSwap and PnPInv+P2P maintain background but lacks capaticy for subject swapping. SwapAnything causes shape mismatch frequently. By contrast, ControlSwap achieves stronger controllability by unifying two aspects: (1) **personalization**, preserving concept identity while adapting it to novel poses and scenes, and (2) **background fidelity**, maintaining background regions without spillover edits. Across challenging cases with large shape or context discrepancies, these qualitative comparisons demonstrate the effectiveness of our ControlLoss and Identifier in addressing the dual challenge of personalization and background preservation.

Table 1: Quantitative comparison on DreamBooth and SwapBench datasets. Best results are in **bold**, second best are underlined.

| Method | Foreground | Background | | | | Overall |
|---|---|---|---|---|---|---|
| | CLIP-I ↑ | PSNR ↑ | LPIPS ↓ | MSE ↓ | SSIM ↑ | CLIP-T (all) ↑ |
| SDS | 0.6118 | 20.77 | 0.3088 | 0.0109 | 0.7609 | 0.2650 |
| DDS | 0.6043 | 24.64 | 0.0907 | 0.0055 | 0.8240 | 0.2895 |
| CDS | 0.6312 | 18.46 | 0.2526 | 0.0350 | 0.7249 | **0.2999** |
| PnPInv+P2P | 0.5508 | 23.16 | 0.2153 | 0.0320 | 0.7517 | 0.2183 |
| PhotoSwap | 0.5638 | 29.18 | 0.0827 | 0.0039 | 0.8385 | 0.2187 |
| SwapAnything | 0.6377 | 29.52 | 0.2030 | 0.0318 | 0.8096 | 0.2802 |
| InstantSwap | 0.6588 | 29.75 | 0.0587 | 0.0030 | 0.8478 | 0.2963 |
| **ControlSwap (ours)** | **0.6632** | **33.73** | **0.0507** | **0.0019** | **0.9291** | 0.2959 |

### 5.2 QUANTITATIVE RESULTS

We also conducted quantitative evaluations on the same methods used in the qualitative comparision. Evaluation metrics include: (1) Foreground CLIP-I (Radford et al., 2021; Hessel et al., 2021; Ruiz et al., 2023) similarity between generated and target subjects, (2) Background PSNR (Horé & Ziou, 2010), LPIPS (Zhang et al., 2018), MSE, and SSIM (Wang et al., 2004) relative to the source background, and (3) Overall CLIP-T similarity to the target prompt.

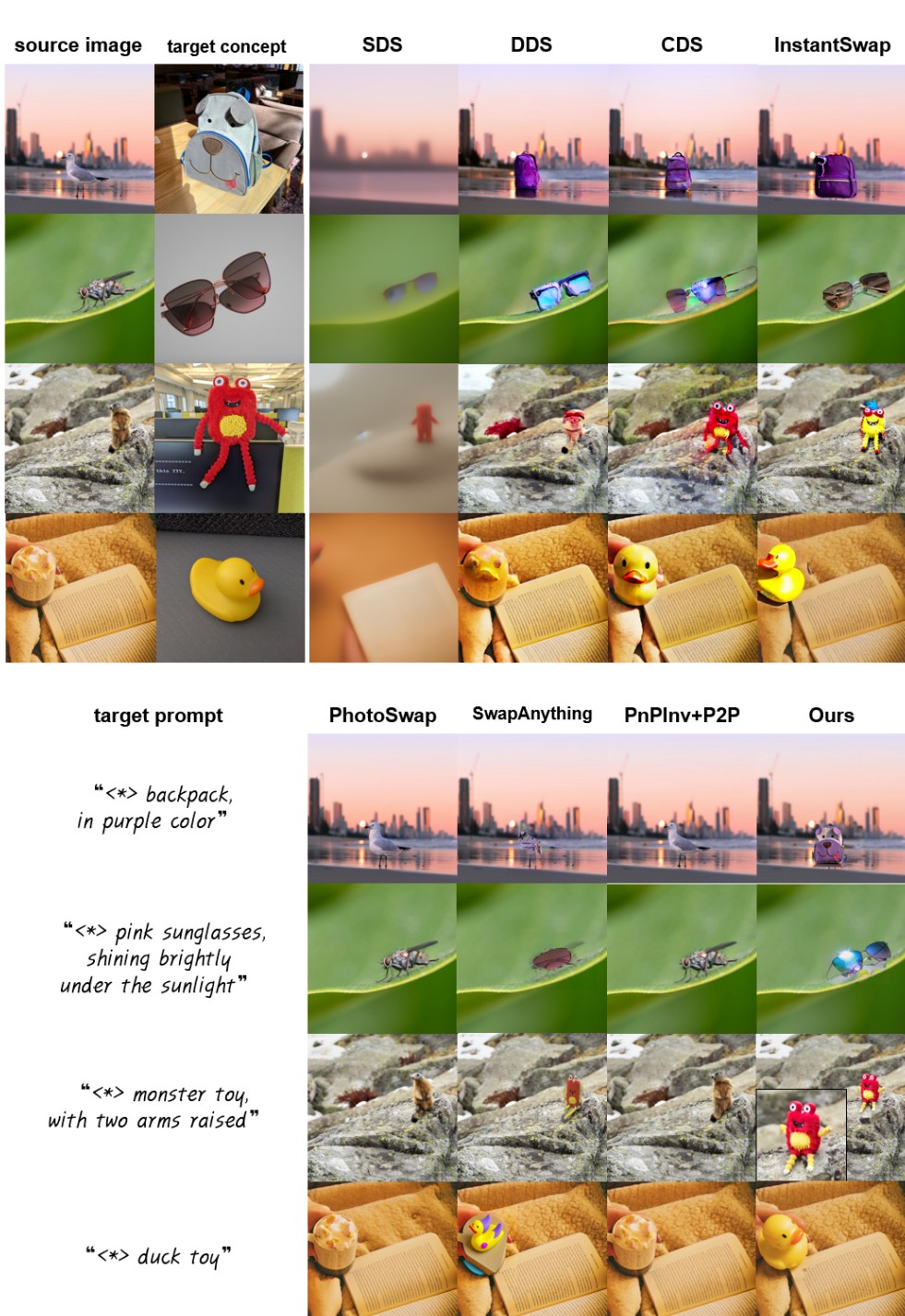

Figure 4: Qualitative comparison across existing methods and our method. Top panel and bottom panel each contains source image, target concept, target prompt, and results of existing methods and ControlSwap (ours). The masks used for these examples are specified in Appendix Fig. 8. ControlSwap achieves better controllability, containing personalization and background fidelity.

As summarized in Table 1, ControlSwap consistently achieves the best balance between controllability and preservation across all metrics. Higher CLIP-I scores demonstrate improved subject personalization to capture identity fidelity. While the CLIP-T scores are comparable, the other methods struggle to preserve concept identity. Meanwhile, background comparision confirm that ControlSwap avoids undesired scene alterations, unlike prior methods that either overfit or underfit to the subject. These results quantitatively validate the effectiveness of our region-aware ControlLoss and Identifiner redesign in unifying subject controllability with background fidelity.

### 5.3 ABLATION STUDY

**Step-Split Optimization**  We treat the transition point between Phase 1 and Phase 2 as same setting with Figure 2 and sweep it across the range $[0, 1150]$ (Fig. 5). The panel shows the source image, target concept, and representative ablations at several split points. Early splits keep the optimization overly tied to the source latent and limit adaptation, whereas very late splits improve personalization but harm visual integration. Therefore, we identify **550 steps** as the optimal split, which removes dependence on the source latent while yielding natural personalization.

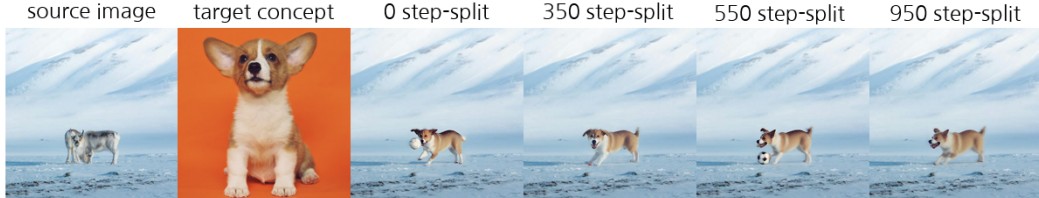

Figure 5: Step-split ablation. We varied the Phase 1 to Phase 2 transition point over the range $[0, 1150]$. Left→right: source image, target concept, and results at various split points. From this sweep, a **550-step** split best balances source-independence and natural personalization.

**Loss Function Graph**  Figure 6 compares results from a standard, object-agnostic DDS-based loss with our proposed region-aware ControlLoss. While the baseline applies uniform guidance over the entire subject mask—treating object-core and near-context pixels equivalently—this often leads to unstable updates, identity drift, or excessive alteration of the surrounding context. In contrast, our ControlLoss explicitly disentangles object and nonobject regions: it promotes identity-preserving transformations within the object core, while enforcing consistency in the near-context and leaving the background protected. This targeted supervision stabilizes optimization, reduces training fluctuations, and produces more faithful subject adaptation alongside robust background preservation.

**Refiner-to-Identifiner effect**  The vanilla refiner denoises the entire latent after re-noising and is not trained to preserve the target concept's identity. As its strength increases (Fig. 7, bottom left), identity drift and background changes emerge from smoothing to a complete change. In contrast, the *identifiner* operates in an inpainting style: it updates only the subject region while freezing background latents. When applied to the target image (top right) or to the source latents (bottom right), it maintains identity and keeps the scene intact even at higher strengths. Extremely large strengths can still nudge global layout; in practice we use $s=0.3$.

## 6 CONCLUSION

We have introduced *ControlSwap*, a controllable personalized subject swapping framework built upon SDXL with an *identifiner* architecture explicitly designed for identity preservation. By adopting ControlLoss that separates object areas, our method addresses two long-standing challenges in subject swapping: (1) target concept personalization, and (2) maintain background regions. Extensive experiments show that ControlSwap achieves superior controllability, producing target-consistent subjects while preserving scene fidelity. These results highlight the value of region-aware design in diffusion-based editing. In future work, we plan to extend our framework to video subject swapping for multi-image batch processing.

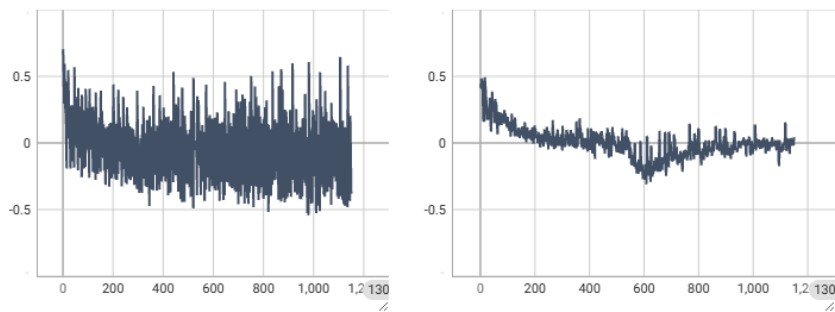

Figure 6: Ablation study on the loss function. DDS Loss (left) applies uniform updates across the subject mask, often causing identity drift. In contrast, our region-aware **ControlLoss** (right) disentangles object-core and nonobject regions, yielding faithful subject personalization while maintaining background fidelity.

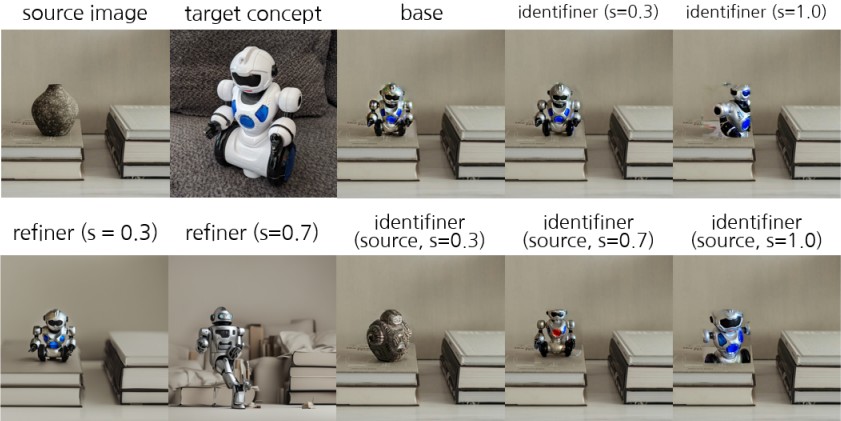

Figure 7: Refiner vs. Identifier. **T**op row — source image, target concept, identifier at representative strengths. Bottom row — Refiner at various strengths, identifier with the source image at multiple strengths. We applied various strength over a range and settings for brevity.

## ETHICS STATEMENT

We study controllable personalized subject swapping for legitimate uses (e.g., creative design, VFX, privacy-preserving augmentation). Experiments conducted on public datasets, and no personal or sensitive data were used. The method is not intended for deceptive or harmful use—non-consensual imagery and political misinformation are explicitly discouraged. We recommend responsible deployment in line with established guidelines from IEEE, UNESCO, and ACM.

## REPRODUCIBILITY STATEMENT

Our implementation details are provided in Section Experiments and Appendix Implementation, and the quantitative evaluation metrics are specified in Section Quantitative Results. As noted in the Section Ethics Statement, all datasets used are publicly available. The implementation code is included in the supplementary materials. To support reproducibility, we will release code, checkpoints, and evaluation scrips upon publication.

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

# A   IMPLEMENTATION

We built our system by integrating several publicly available implementations. We primarily followed the default settings. All experiments were conducted on an NVIDIA RTX A6000 GPU.

**Pretrained DreamBooth Models.**   We used the HuggingFace diffusers repository[1] for Dream-Booth fine-tuning (Ruiz et al., 2023). Specifically, we employed DreamBooth [2] for Stable Diffusion backbone and DreamBooth LoRA SDXL [3] for SDXL backbone. All models were trained with mixed precision fp16, gradient accumulation 4, batch size 1, learning rate $1 \times 10^{-4}$, and 500 training steps.

**SDS and DDS.**   The Score Distillation Sampling (Poole et al., 2023) and Denoising Diffusion Score (Hertz et al., 2023a) objectives were implemented following Google's Prompt-to-Prompt notebook[4]. We reproduced both of the default setting with cut loss weight coefficient 3.0, dds loss weight coefficient 1.0, guidance scale 7.5, learning rate 0.1, and 200 iteration steps.

**CDS.**   We employed CDS (Nam et al., 2024) objective from the official repository[5]. We used default settings as guidance scale 7.5, learning rate 0.1, and 200 iteration steps.

**InstantSwap.**   We used official InstantSwap (Zhu et al., 2025) [6] implementation. We used the default setting as SSGU factor 5, guidance scale 7.5, learning rate 0.1, and 550 iteration steps.

**PhotoSwap.**   PhotoSwap Implementation was adapted from the official PhotoSwap (Gu et al., 2023) implementation[7], which provides a PyTorch-based pipeline for null-text and DDIM inversion (Mokady et al., 2023). We followed their default hyperparameters, using 50 DDIM steps for inversion, cross map replace parameter 0.6, self output replace parameter 0.8, self map replace parameter 0.0, guidance scale 7.5.

**SwapAnything.**   We referenced the official SwapAnything (Gu et al., 2024) framework[8]. We applied semgentation mask by using (Kirillov et al., 2023), followed its default setting as 50 DDIM steps for inversion, total diffusion steps 50, cross map replace parameter [0.1, 0.3, 0.5, 0.7], self output replace parameter [0.1, 0.3, 0.5, 0.7], self map replace parameter 0.0, and guidance scale 7.5.

**PnPInversion.**   We used official implementation of PnPInversion[9]. We used the default parameter as cross replace steps parameter 0.4, self replace steps parameter 0.6, learning rate 0.1, and guidance scale 7.5.

**ControlSwap**   We adopted (Hertz et al., 2023a) as our baseline and modified the code to fit our method. Following (Zhu et al., 2025), we set the SSGU factor to 3. We empirically set the step split to 550 out of a total of 1150 steps. The object mask was obtained using (Kirillov et al., 2023). We employed the Lookahead optimizer with a learning rate of 0.1. The guidance scale was set to 10.0 for the base model and 7.5 for the refiner. The refiner strength was fixed to 0.3.

Table 2: Per-image inference time (mm:ss).

(a) SDS / DDS / CDS / InstantSwap

| Method | SDS | DDS | CDS | InstantSwap |
|---|---|---|---|---|
| **time (mm:ss)** | 00:18 | 00:23 | 00:35 | 00:34 |

(b) PhotoSwap / SwapAnything / PnPInv+P2P / ControlSwap

| Method | PhotoSwap | SwapAnything | PnPInv+P2P | ControlSwap (ours) |
|---|---|---|---|---|
| **time (mm:ss)** | 01:25 | 01:24 | 01:25 | 02:16 |

## B  INFERENCE TIME

We measured the per-image inference time for SDS (Poole et al., 2023), DDS (Hertz et al., 2023a), CDS (Nam et al., 2024), InstantSwap (Zhu et al., 2025), PhotoSwap (Gu et al., 2023), SwapAnything (Gu et al., 2024), and PnPInv+P2P (Ju et al., 2024; Hertz et al., 2023b), alongside **Control-Swap (ours)**. In general, runtimes are between 0 and 2 minutes. Notably, PhotoSwap and SwapAnything spent approximately $1\min 16\,s$ to $17\,s$ on DDIM inversion, while the subsequent forward pass took about $8\,s$. Our method required $2\min 15\,s$ for the base model and $1\,s$ for the identifier. Because prior work did not account for controllability, our method entails an inherently higher computational cost. Empirically, this additional cost is accompanied by consistently improved results.

## C  LARGE LANGUAGE MODELS (LLMS) USAGE STATEMENT

We used OpenAI's ChatGPT[10] for limited writing assistance: (i) recommending diverse positive prompts, (ii) refining English word choices from a native-speaker's perspective to improve clarity, and (iii) summarizing a public survey (Huang et al., 2025) to study background knowledge on editing efficiently. Technical ideas, methodology, experiments, analyses, and final claims were produced and verified by the authors.

## D  MASKS USED FOR FIG. 4

Fig. 8 visualizes the binary masks used in Fig. 4. The foreground object mask is obtained by applying (Kirillov et al., 2023) to the source image within the provided bounding box to extract the object shape. The foreground nonobject mask is defined as the region inside the bounding box excluding the object mask. Bounding boxes are provided by the SwapBench subset of ConSwapBench.

---

[1] https://github.com/huggingface/diffusers

[2] https://github.com/huggingface/diffusers/blob/main/examples/dreambooth/train_dreambooth.py

[3] https://github.com/huggingface/diffusers/blob/main/examples/dreambooth/train_dreambooth_lora_sdxl.py

[4] https://github.com/google/prompt-to-prompt/blob/main/DDS_zeroshot.ipynb

[5] https://github.com/HyelinNAM/ContrastiveDenoisingScore

[6] https://github.com/chenyangzhu1/InstantSwap

[7] https://github.com/eric-ai-lab/photoswap

[8] https://github.com/eric-ai-lab/swap-anything

[9] https://github.com/cure-lab/PnPInversion

[10] https://chatgpt.com/

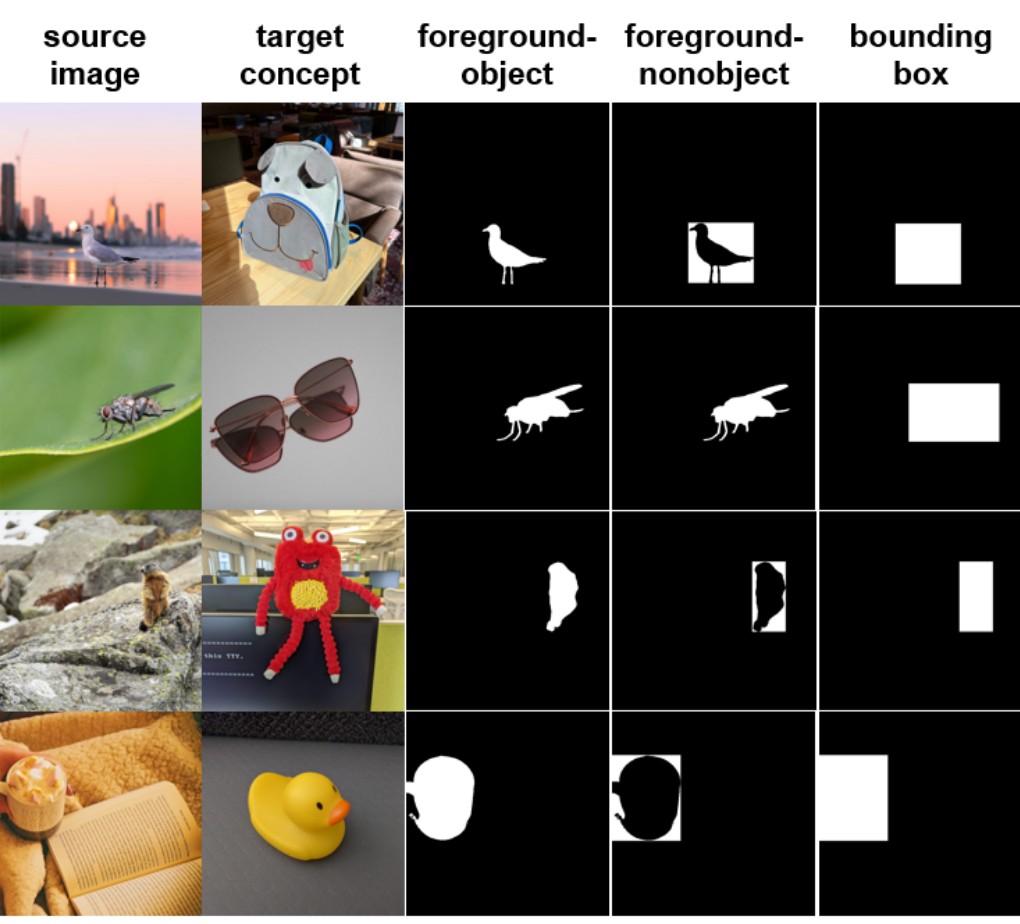

Figure 8: Masks used for Fig. 4. Columns (left→right): source image, target concept image, foreground-object mask, foreground-nonobject mask, and bounding box mask.

