# OpenReview forum: "ControlSwap: Controllable Personalized Subject Swapping"
_ICLR.cc/2026/Conference — ICLR 2026 Conference Withdrawn Submission_

### Official Review · Reviewer_8yfP · 2025-10-18

**Soundness:** 2
**Presentation:** 3
**Contribution:** 1
**Rating:** 2
**Confidence:** 4

**Summary:**

An SDXL-based editing pipeline for subject swapping. The refiner is turned into an “Identifiner” for identity preservation, and a region-aware ControlLoss separates object vs. non-object optimization to avoid background drift. Shown on DreamBooth identities and SwapBench images.

**Strengths:**

1. Addresses a real failure mode (background changes).
2. Compatible with SDXL stack.

**Weaknesses:**

1. Minimal algorithmic innovation.
2. Lacks strong quantitative comparisons (e.g., InstantID, recent swap/edit methods).
3. No discussion of safety or misuse.

**Questions:**

1. How are masks obtained (SAM? manual?) and how robust is the method to mask errors?
2. Can it handle multi-subject scenes or video sequences?
3. Any quantitative metric for identity preservation vs. background fidelity?

**Details Of Ethics Concerns:**

No discussion of safety or misuse.

---

### Official Review · Reviewer_FKMM · 2025-10-30

**Soundness:** 3
**Presentation:** 3
**Contribution:** 3
**Rating:** 4
**Confidence:** 4

**Summary:**

This paper presents ControlSwap, which is a controllable framework for personalized subject swapping built on Stable Diffusion XL. It replaces the subject in an image with a target identity while keeping the background intact. The model uses an Identifiner to preserve identity and a region-aware loss to balance subject editing with background protection. Experiments show clear improvements in controllability and visual fidelity over prior methods.

**Strengths:**

1. The qualitative and quantitative results demonstrate consistent gains in identity fidelity and background realism across multiple benchmarks.
2. The ControlLoss divides the image into object, surrounding, and background areas, allowing the model to edit the subject more precisely without disturbing the surrounding scene.
3. The paper is easy to follow with clear presentation.

**Weaknesses:**

1. The core idea builds upon existing diffusion and inpainting-based editing frameworks. While the Identifiner is well implemented, it is not a novel method, which has appeared in many previous works.
2. he paper relies mainly on automatic metrics like CLIP similarity and PSNR, which may not fully reflect perceptual quality or realism. A human study would provide stronger evidence for visual improvement.

**Questions:**

Since controllability and realism are perceptual qualities, have the authors conducted or considered user studies to validate the quantitative metrics? If not, how confident are they that CLIP-based metrics correlate well with human perception?

---

### Official Review · Reviewer_Taes · 2025-10-31

**Soundness:** 1
**Presentation:** 1
**Contribution:** 1
**Rating:** 2
**Confidence:** 5

**Summary:**

This paper proposes ControlSwap, an SD-XL framework for image editing, especially object swap. It proposes ControlLoss for optimizing the foreground and background separately.

**Strengths:**

1. The motivation for ControlLoss is sound, as it optimizes the foreground and background separately. The experiments also validate its effectiveness.

2. The two-phase pipeline seems intuitive: it first identifies the editing and unedited areas, then integrates the subject.

**Weaknesses:**

1. The paper's writing quality is extremely poor. Figure 2,3 is not a vector graphic and cannot be zoomed in on (or "cannot be viewed at a larger size"). Additionally, there is an obvious typo on line 199 after the word "background". This level of writing is not suitable for submission to ICLR.

2. The base model is SD-XL and it uses DDPM sampling, which is outdated. Commonly, in T2I generation, researchers now use models like FLUX with flow matching. The proposed ControlSwap method is not applicable to these more recent T2I models.

**Questions:**

See weakness

---

### Note · Authors · 2025-11-14

I have read and agree with the venue's withdrawal policy on behalf of myself and my co-authors.